# Fetoscopic endoluminal tracheal occlusion with Smart-TO balloon: Study protocol to evaluate effectiveness and safety of non-invasive removal

Nicolas Sananès[1,2☯]*, David Basurto[3☯], Anne-Gaël Cordier[4], Caroline Elie[5], Francesca Maria Russo[3,6], Alexandra Benachi[4‡], Jan Deprest[3,6,7‡]

**1** Department of Maternal Fetal Medicine, Strasbourg University Hospital, Strasbourg, France, **2** INSERM 1121 'Biomaterials and Bioengineering', Strasbourg University, Strasbourg, France, **3** MyFetUZ Fetal Research Center, Department of Development and Regeneration, Cluster Woman and Child, Biomedical Sciences, KU Leuven, Leuven, Belgium, **4** Department of Maternal fetal Medicine, Antoine–Béclère Hospital - Paris–Saclay University, Clamart, France, **5** Clinical Research Unit/Clinical Investigation Center, Necker-Enfants Malades Hospital, Paris, France, **6** Clinical Department of Obstetrics and Gynaecology, University Hospitals Leuven, Leuven, Belgium, **7** Institute for Women's Health, University College London, London, United Kingdom

☯ These authors contributed equally to this work.
‡ AB and JD also contributed equally to this work. shared last authorship.
* Nicolas.sananes@chru-strasbourg.fr

**Data Availability Statement:** No datasets were generated or analysed during the current study. All

## Abstract

### Introduction

One of the drawbacks of fetoscopic endoluminal tracheal occlusion (FETO) for congenital diaphragmatic hernia is the need for a second invasive intervention to reestablish airway patency. The "Smart-TO" (Strasbourg University-BSMTI, France) is a new balloon for FETO, which spontaneously deflates when positioned near a strong magnetic field, e.g., generated by a magnetic resonance image (MRI) scanner. Translational experiments have demonstrated its efficacy and safety. We will now use the Smart-TO balloon for the first time in humans. Our main objective is to evaluate the effectiveness of prenatal deflation of the balloon by the magnetic field generated by an MRI scanner.

### Material and methods

These studies were first in human (patients) trials conducted in the fetal medicine units of Antoine–Béclère Hospital, France, and UZ Leuven, Belgium. Conceived in parallel, protocols were amended by the local Ethics Committees, resulting in some minor differences. These trials were single-arm interventional feasibility studies. Twenty (France) and 25 (Belgium) participants will have FETO with the Smart-TO balloon. Balloon deflation will be scheduled at 34 weeks or earlier if clinically required. The primary endpoint is the successful deflation of the Smart-TO balloon after exposure to the magnetic field of an MRI. The secondary objective is to report on the safety of the balloon. The percentage of fetuses in whom the balloon is deflated after exposure will be calculated with its 95% confidence interval.

relevant data from this study will be made available upon study completion.

**Funding:** DB is funded by the Erasmus + Programme of the European Union (Framework Agreement number: 2013-0040). This publication reflects the views only of the authors, and the Commission cannot be held responsible for any use which may be made of the information contained therein. JD is funded by the Great Ormond Street Hospital Charity Fund. In Paris, the present clinical protocol is funded by a grant from Assistance Publique – Hôpitaux de Paris (CRC19). In Leuven, the study is funded by the Klinisch Onderzoeks- en Ontwikkelingsfonds of the UZ Leuven (S65423). The funders had and will not have a role in study design, data collection and analysis, decision to publish, or preparation of the manuscript.

**Competing interests:** 'Nicolas Sananès is the primary co-inventor of the Smart-TO balloon. None of the authors has any financial interest in BS-Medical Tech Industry, manufacturing the balloon. There are no other conflicts of interest. This does not alter our adherence to PLOS ONE policies on sharing data and materials.

Safety will be evaluated by reporting the nature, number, and percentage of serious unexpected or adverse reactions.

## Conclusion

These first in human (patients) trials may provide the first evidence of the potential to reverse the occlusion by Smart-TO and free the airways non-invasively, as well a safety data.

## Introduction

### Background

Congenital diaphragmatic hernia (CDH) is a birth defect characterized by failed closure of the diaphragm. This enables abdominal viscera to herniate into the thoracic cavity, leading to hypoplastic lungs and impaired lung vasculature [1]. Fetoscopic Endoluminal Tracheal Occlusion (FETO) increases fetal lung volume and therefore can improve survival in selected fetuses with CDH. Recently two parallel randomized controlled trials in fetuses with isolated left-sided CDH with severe and moderate pulmonary hypoplasia respectively were concluded [2, 3]. In severe hypoplasia the balloon was inserted early ($27^{+0}$ to $29^{+6}$ weeks' gestation) and FETO improved survival from 15% to 40% (Table 1) [3]. A comparable improvement in survival (20% to 42%) was achieved in fetuses with severe right-sided CDH [4]. In moderate hypoplasia, the balloon was inserted later ($30^{+0}$ to $31^{+6}$ weeks' gestation) in an effort to reduce the risks of very preterm birth. In that study, FETO improved survival from 50% to 63%, but the difference in survival was not statistically significant [2]. Analysis of the pooled data from the two randomized trials demonstrated that FETO increases survival in both severe and moderate disease (Table 1), but the observed lesser effect in the moderate group is most likely a mere consequence of the delayed insertion of the balloon in moderate hypoplasia [5].

An adverse side-effect of FETO is that it increases the risk for iatrogenic preterm membrane rupture and preterm birth [6]. In the TOTAL trials, that risk was inversely related to the gestational age at the insertion of the balloon [5]. Although the trials did not demonstrate any obvious differences between the FETO and control groups in prematurity-related complications, they were not powered to study differences in these secondary outcomes. Long-term outcomes will have to further elucidate that, but it would seem logical to expect a measurable effect of prematurity when large numbers are available.

Another disadvantage of the current procedure is the need for an invasive, second intervention to reverse the occlusion and re-establish airway patency. Balloon removal is scheduled electively at 34 weeks, or earlier if required. Reversal of the occlusion is preferentially performed at least 24 hours before birth, as that seems associated with an increased survival [2, 7–9]. Reversal is at present an invasive procedure that can be performed prenatally by either ultrasound-guided puncture, fetoscopy, or, less ideal, after delivery of the baby prior to cord clamping while the fetus is maintained on placental circulation or after the cord is clamped at birth after vaginal delivery [9]. Airway re-establishment requires a specialist team familiar with the procedure and that is available 24/7 [9]. In a large series, 28% of balloon removals were in an emergency setting [9]. The only neonatal deaths that occurred, were when balloon reversal was attempted in centers without experience or that were unprepared [9]. Even in experienced centers balloon removal can fail, as observed in the TOTAL trial [2]. Also, patients may be

**Table 1. Outcomes of fetuses diagnosed with isolated congenital diaphragmatic hernia (CDH) in the prenatal period, either left or right-sided, expectantly managed during pregnancy or having tracheal occlusion within the "Tracheal Occlusion to Accelerate Lung Growth" (TOTAL) trial or and a large study on right-sided CDH under the same management protocol.**

| Side, severity | Criteria severity on ultrasound | Survival to discharge | | RR (95% CI) |
|---|---|---|---|---|
| | | Expectant | FETO | |
| **Isolated left sided CDH–TOTAL trials** | | | | |
| **TOTAL** *severe* [3] | O/E LHR <25.0% Irrespective of liver position | 6/40 (15%) | 16/40 (40%) | **2.67 (1.22–6.11)** |
| **TOTAL** *moderate* [2] | O/E LHR 25.0–34.9%, any liver position O/E LHR 35.0–44.9% & liver into chest | 49/98 (50%) | 62/98 (63%) | **1.27 (0.99–1.63)** |
| **Isolated left sided CDH–Pooled analysis TOTAL data** | | | | |
| **Late insertion** | O/E LHR 0.0–34.9%, any liver position O/E LHR 35.0–44.9% & liver into chest | 55/142 (39%) | 79/145 (54%) | **A OR: 1.78 (1.05–3.01)** |
| **Early insertion** | O/E LHR 0.0–34.9%, any liver position O/E LHR 35.0–44.9% & liver into chest | | | **A OR: 2.73 (1.15–6.49)** |
| **Isolated right sided CDH** | | | | |
| *Severe* [4] | O/E LHR <50% Irrespective of liver position | 7/34 (20%) | 53/125 (42%) | **2.84 (1.15–7.01)** |

Abbreviations: RR, risk ratio; CI, confidence interval; O/E LHR, observed-to-expected lung-to-head ratio; A, adjusted; OR, odds ratio.

non-compliant and move away from the fetal surgery center [2]. The second procedure inherently adds risks for the mother and fetus. These can be directly procedure-related, but also indirectly, by increasing the risk for membrane rupture later on [9]. In conclusion, the occlusion period is a serious burden on *patients* who are requested to stay close to the FETO center until balloon removal, as well as for the *fetal surgery centers* because of the need for permanently available staff. All these conditions, limit the acceptability of FETO as being practiced today.

The University of Strasbourg, France, in partnership with BS Medical Tech Industry (BS-MTI), Niederroedern, France, developed an alternative occlusion device, referred to as "Smart-TO" [10]. Compared to the currently used Goldbal2® (Balt, Montmorency, France) balloon, the Smart-TO balloon has identical dimensions in its inflated state and is made of the same material (latex). Around the balloon neck, there is a metallic cylinder and inside a magnetic ball, which together act as a valve. Deflation occurs under the influence of a strong magnetic field, which is present around any clinical MRI machine. For that, it is sufficient for the pregnant woman to walk around the MRI machine. This enables non-invasive, externally controlled balloon deflation. Therefore, the Smart-TO balloon may address issues related to the unplug procedure, i.e. neonatal deaths by failure of balloon removal, morbidity related to a second fetal surgery procedure, need for FETO centers with experienced team available 24/7, and need for the pregnant women to stay close to a FETO center during the whole duration of the occlusion.

The Smart-TO balloon been tested preclinically by BS-MTI (the manufacturer), University of Strasbourg, Simian Laboratory Europe and the KU Leuven. In-vitro tests including permeability, occlusion, and deflation in a simulated environment were performed by BS-MTI (unpublished data). Deflation tests were performed using a mannequin in a simulated "in-utero" environment, with the fetus and the mother in different positions and heights [11]. In that experiment, deflation was successfully achieved using a 1.5T MRI in 100% of cases in a maternal standing position as well as when the maternal position was 'lying on a stretcher'. The only case of failure occurred when the maternal position was 'sitting in a wheelchair', likely because of the distance between the MRI scanner and the patient in this scenario.

In vivo animal tests included the demonstration of similar lung growth and short-term tracheal side effects as the Goldbal 2 balloon in fetal lambs [11, 12]. In the latter experiment, fetal lambs expelled the Smart-TO balloon following exposure to the fringe field of a 3T MRI. Finally, feasibility of balloon insertion, persisting occlusion until reversal, and spontaneous expulsion of the Smart-TO balloon was confirmed in non-human primates [10]. Therefore, this novel medical device should now be evaluated in a first in human (patients) trial. For that purpose we designed two studies, one at Antoine–Béclère Hospital Paris–Saclay University, Clamart, France referred to as "Smart-FETO", and one at the University Hospitals Leuven (UZ Leuven), Belgium, referred to as "Smart-Removal". Conceived in parallel, protocols were amended by the local Ethics Committee on Clinical Studies or its equivalent, resulting in a limited number of differences (Table 2).

## Objectives and hypotheses

The main objective of these studies is to demonstrate the ability to consistently deflate the balloon prenatally by the magnetic fringe field generated by a clinical MRI scanner, and that it will be expelled from the airways. Secondary objective is to report on the safety of the balloon. We hypothesize that there will not be any serious adverse effects directly related to the Smart-TO balloon itself. Other objectives include assessment of prematurity, preterm premature rupture of membranes, lung growth, neonatal survival, and the need for oxygen supplementation at discharge from the hospital.

## Methods

### Design plan

**Study type.**   These clinical trials are single-arms interventional feasibility studies. Eligible consecutive consenting women will have FETO with the Smart-TO balloon.

**Setting.**   These trials are conducted at two centers i.e., the Antoine–Béclère Hospital—Paris–Saclay University, Clamart, France, and the University Hospitals Leuven, Belgium.

### Sampling plan

**Existing data.**   Both trials have been registered prior to their inception (ClincalTrial.gov NCT04931212 and NCT05100693). The first inclusion in France was on August 4[th], 2021, and in Belgium on September 10[th], 2021.

**Recruitment.**   Recruitment of participants will be at the latest one day before planned balloon placement. Written informed consent will be obtained from all participants.

*Inclusion criteria.*

- Patient aged 18 years or more and who can consent,

- Singleton pregnancy with a fetus with an isolated congenital diaphragmatic hernia (i.e., no additional major structural malformation nor genetic abnormality)

- -Eligible for FETO, i.e. having severe pulmonary hypoplasia defined as, in left-sided cases, an observed-to-expected 'lung-to-head ratio' (O/E LHR) <25% irrespective of the liver position, or moderate pulmonary hypoplasia defined as O/E LHR 25–34.9% irrespective of the liver position or O/E LHR 35–44.9% with liver herniation, and, in UZ Leuven, fetuses with right-sided CDH with severe hypoplasia (O/E LHR < 50%). The O/E LHR measurement can be performed either by trace or anteroposterior diameters of the contralateral lung.

**Table 2. Inclusion criteria and outcome measurements in both studies.** Differences are displayed in bold.

| Smart-FETO (Paris) | Smart-Removal (Leuven) |
|---|---|
| Inclusion criteria | |
| • Patient can consent, has 18 years or more, and **has medical insurance** | • Patient can consent, has 18 years or more |
| • Singleton pregnancy with a fetus with an isolated left CDH with severe or moderate lung hypoplasia | • Singleton pregnancy with a fetus with an isolated left CDH with severe or moderate lung hypoplasia or **right CDH with severe lung hypoplasia** |
| Primary endpoint | |
| • Balloon deflation rate after MRI exposure (by ultrasound) and **balloon expulsion from the fetal airways (by postnatal chest X•ray)** | • Balloon deflation rate after MRI exposure (by ultrasound). |
| Secondary endpoints | |
| Prenatal | Prenatal |
| • Spontaneous balloon deflation prior to MRI exposure (by ultrasound) | • Spontaneous balloon deflation prior to MRI exposure (by ultrasound). |
| • Lung growth (O/E LHR) measured by ultrasound before unplugging. | • Lung growth **2 weeks after FETO** (O/E LHR). |
| • Presence of the inflated balloon in the trachea before the unplugging procedure, assessed by a test ultrasound just before the patient circuits the MRI device (position and **height and width** of the balloon). | • Gestational age at membrane rupture |
| Postnatal: | Postnatal: |
| • Prematurity before 32, 34 and 37 weeks of amenorrhoea | • Gestational age at delivery |
| | • Balloon expulsion from the fetal airways (by postnatal chest X•ray) |
| • **Localisation of the balloon by postnatal chest X•ray of the newborn** | • **Localisation of the balloon either by (1) direct visualization within the amniotic fluid, membranes or placenta, (2) postnatal chest X-ray of the newborn, and (3) ultrasound of the postpartum uterus.** |
| Neonatal: | Neonatal: |
| • Survival at discharge from the hosptital | • Survival at discharge from the NICU |
| • Survival at 6 months | • **Tracheal diameter on first postnatal chest X-ray** |
| • Need for oxygen supplementation at 6 months | • **Assessment for any local side effects of the balloon (signs or symptoms of tracheomegaly and /or tracheomalacia)** |
| Adverse events: | Adverse events: |
| • Any adverse event, either in the mother or the fetus or newborn, at whatever time point between insertion and discharge from the neonatal unit | • Any adverse event, either in the mother or the fetus or newborn, at whatever time point between insertion and discharge from the neonatal unit |
| Sample size | |
| **n = 20. If efficacy is 100%, leading to a lower boundary of 95% CI equal to 83%.** | **n = 25 for a 95% CI with a lower boundary of 85% [13] and an expected loss rate of 8% (n = 2) due to the need for removal on placental circulation or spontaneous balloon deflation.** |

Abbreviations: CDH, congenital diaphragmatic hernia; MRI, magnetic resonance image; O/E LHR, observed to expected lung to head ratio; CI, confidence interval.

- At a stage of pregnancy compatible with inserting the balloon of between 27 and $29^{+6}$ WA for severe hernias and between 30 and $31^{+6}$ WA for moderate hernias in France.

    *Exclusion criteria.*

- Maternal contraindication to fetoscopy

- Preterm premature rupture of the membranes (PPROM) or any condition strongly predisposing to PPROM or premature delivery

- Patient does not consent to stay close to the FETO center during the occlusion period.

**Sample size.** Independent sample size calculation has been performed in both centers. In Paris (France), we hypothesized a 100% deflation and expulsion rate. If this hypothesis is effectively verified on 20 patients, we can then say that the probability of the balloon deflating with

the MRI is of 100% with a 95% confidence interval (CI) between 83 and 100% (calculation performed with the exact method). In Leuven (Belgium), the estimated number is 23 patients, in order to achieve a 95% CI with a lower boundary of 85% [13]. The theoretical possibility of spontaneous balloon deflation, or the impossibility to expose the patient to MRI at the time of balloon removal (e.g., in an emergency requiring removal on placental circulation) was considered as possible (n = 2), so that a total of 25 patients are to be recruited.

## Variables

**Measured variables.** These include administrative data, data on the index pregnancy, characteristics of the fetus, on the FETO procedure, follow-up ultrasound measurements, balloon removal, delivery, and the neonatal follow-up period until discharge from the neonatal intensive care unit (NICU) (Table 3).

**Primary endpoint.** In France, the primary endpoint is the successful deflation of the Smart-TO balloon after exposure to the fringe field of the MRI, assessed through ultrasound immediately after MRI exposure and the expulsion of the Smart-TO balloon from the airways, as documented by a X-ray of the neonatal chest at birth (the valve of the balloon is radio-opaque).

In Leuven, only the successful deflation of the Smart-TO balloon after exposure to the fringe field of the MRI is required for efficacy; this endpoint will be then considered as the common primary endpoint.

For Belgium, the expulsion of the Smart-TO balloon from the airways is considered as a secondary endpoint.

**Secondary endpoints.** Secondary endpoints are displayed in Table 2.

## Statistical analysis plan

The percentage of fetuses in whom the balloon deflated at exposure and fetuses that expelled the balloon from the fetal airways will be calculated with its 95% confidence interval using the binomial method [13].

Safety will be evaluated by reporting the nature, number, and percentage of serious unexpected or adverse reactions. Other secondary endpoints will be described. Quantitative data will be expressed as median and inter-quartile-range (IQR), qualitative data will be expressed as numbers and percentages.

There will be no imputation of missing data for secondary outcomes.

## Intervention

The study schedule is displayed in Fig 1.

**FETO.** The FETO procedure will be performed as earlier described [14]. Regarding the Smart TO use:

- The catheter system is introduced in the sheath of the endoscope and back loaded with the Smart-TO balloon. The balloon is then tested by inflation with 0.7 mL of sterile saline and deflated with its proper stylet, following which the latter is withdrawn.

- The balloon is positioned between the carina and the vocal cords, inflated with 0.7 mL sterile saline, and detached by the combination of gentle traction of the delivery system and counter pressure with the endoscope.

**Table 3. List of variables from both studies.** Differences are displayed in bold.

| Paris (Smart FETO) | Leuven (Smart Removal) |
|---|---|
| **Administrative data** | |
| • Last name initial | Subject number |
| • First name initial | |
| • Date of birth | Date of birth |
| • Physician responsible for the inclusion | • Physician responsible for the inclusion |
| • Referring center prenatal | • Referring center prenatal |
| • Referring fetal medicine specialist | • Referring fetal medicine specialist |
| • Center postnatal care | • Center postnatal care |
| • Postnatal specialist | • Postnatal specialist |
| **Selection visit** | |
| • Date of selection visit | • Date of selection visit |
| • Have all inclusion criteria been met? yes; no; | • Have all inclusion criteria been met? yes; no; |
| • Date of signature of consent | • Date of signature of consent |
| • Parity | • Parity |
| • Conception: spontaneous; assisted; unknown | • Conception: spontaneous; assisted; unknown |
| • Estimated day of delivery | • Estimated day of delivery |
| • Pre-pregnancy weight (kg) | • Pre-pregnancy weight (kg) |
| • Height (cm) | • Height (cm) |
| • Smoking: 0/d; 1-10/d; 10-20/d; 21 or more/d | • Smoking: 0/d; 1-10/d; 10-20/d; 21 or more/d |
| • Alcohol use: 0; once a week; 2-4/w; >5/w | • Alcohol use: 0; once a week; 2-4/w; >5/w |
| • Drugs: yes; no | • Drugs: yes; no |
| • Concomitant diseases: yes; no; describe | • Concomitant diseases: yes; no; describe |
| • Ethnicity: Caucasian; North African; African; Asian; Other | • Ethnicity: Caucasian; North African; African; Asian; Other |
| • Gestational age | • Gestational age |
| • Severity of hernia: moderate; severe | • Severity of hernia: moderate; severe |
| • **Method for LHR measurement: tracing; longest diameter; anteroposterior diameter and perpendicular** | • **Hernia side; left; right** |
| • O/E LHR (%) | • O/E LHR (%) |
| • Liver herniation: down; up | • Liver herniation: down; up |
| • Grading of stomach position according to Cordier classification: 1; 2; 3; 4 | • Grading of stomach position according to Cordier classification: 1; 2; 3; 4; **N/A (right-CDH)** |
| • Cervical length (mm) | • Cervical length (mm) |
| • Chorionic membrane separation | • Chorionic membrane separation |
| • Deepest vertical amniotic fluid pocket (cm) | • Deepest vertical amniotic fluid pocket (cm) |
| • Placental position: anterior; posterior; fundal | • Placental position: anterior; posterior; fundal |
| • Placenta previa: no; yes | • Placenta previa: no; yes |
| • Karyotype performed: no; yes | • Karyotype performed: no; yes |
| • CGHarray performed: no; yes | • CGHarray performed: no; yes |
| • Results karyotype / CGHarray: normal; abnormal | • Results karyotype / CGHarray: normal; abnormal |
| **Pre-FETO visit** | |
| • Date | • Date |
| • Gestational age | • Gestational age |
| | • Severity of hernia: moderate; severe |
| • **Method for LHR measurement: tracing; longest diameter; anteroposterior diameter and perpendicular** | • **Hernia side; left; right** |
| • O/E LHR (%) | • O/E LHR (%) |
| • Liver herniation: down; up | • Liver herniation: down; up |

*(Continued)*

**Table 3.** (Continued)

| Paris (Smart FETO) | Leuven (Smart Removal) |
|---|---|
| • Grading of stomach position according to Cordier classification: 1; 2; 3; 4 | • Grading of stomach position according to Cordier classification: 1; 2; 3; 4; **N/A (right-CDH)** |
| • Cervical length (mm) | • Cervical length (mm) |
| • Chorionic membrane separation | • Chorionic membrane separation |
| • Deepest vertical amniotic fluid pocket (cm) | • Deepest vertical amniotic fluid pocket (cm) |
| • Placental position: anterior; posterior; fundal | • Placental position: anterior; posterior; fundal |
| • Placenta previa: no; yes | • Placenta previa: no; yes |
| • Date of MRI | • Date of MRI |
| • O/E Total lung volume MRI (%) | • O/E Total lung volume MRI (%) |
| • Additional anomalies on MRI: yes; no; describe | • Additional anomalies on MRI: yes; no; describe |
| **FETO procedure** | |
| • Date | • Date |
| • Operator name | • Operator name |
| • Anesthesia: local; loco-regional (spinal; epidural; spinal-epidural); general | • Anesthesia: local; loco-regional (spinal; epidural; spinal-epidural); general |
| • **Complications related to anesthesia** | |
| • **Initial position of fetus: cephalic, breech, transverse** | |
| • **Fetal version: no; yes** | |
| • **Final position of fetus: cephalic, breech, transverse** | |
| • Complications / difficulties related with balloon preliminary tests: no; yes | • Complications / difficulties related with balloon preliminary tests: no; yes |
| • Complications / difficulties related with connection of balloon with catheter: no; yes | • Complications / difficulties related with connection of balloon with catheter: no; yes |
| • Complications / difficulties related with balloon positioning in working channel: no; yes | • Complications / difficulties related with balloon positioning in working channel: no; yes |
| • Complications / difficulties related with trocar insertion: no; yes | • Complications during FETO procedure: yes; no describe |
| • **Complications / difficulties related with tracheoscopy: no; yes** | |
| • Complications / difficulties related with balloon filling: no; yes | • Complications / difficulties related with balloon filling: no; yes |
| • Complications / difficulties related with balloon withdrawal: no; yes | • Complications / difficulties related with balloon withdrawal: no; yes |
| • Success positioning of Smart-TO balloon: no; yes | • Success positioning of Smart-TO balloon: no; yes |
| • Amniodrainage during FETO: no; yes | • Amniodrainage during FETO: no; yes |
| • Volume of amniodrainage (ml) | • Total in utero time (min) |
| • Total in utero time (min) | • Total in utero time (min) |
| • Time between beginning of fetoscopy and balloon withdrawal | • Time between beginning of fetoscopy and balloon withdrawal |
| • **Tocolysis** | |
| **Postoperative ultrasound** | |
| • Date | • Date |
| • Alive fetus: yes; no | • Alive fetus: yes; no |
| • Balloon in place yes; no | • Balloon in place yes; no |
| • Balloon length (mm) | • Balloon length (mm) |
| • Balloon width (mm) | • Balloon width (mm) |
| • Cervical length (mm) | • Cervical length (mm) |
| • Chorionic membrane separation | • Chorionic membrane separation |
| • Deepest vertical amniotic fluid pocket (cm) | • Deepest vertical amniotic fluid pocket (cm) |
| • Any additional findings: yes; no; describe | • Any additional findings: yes; no; describe |
| • Did any A/E/SAE occur? Yes, no; describe | • Did any A/E/SAE occur? Yes, no; describe |
| Did any device deficiency occur? Yes, no; describe | Did any device deficiency occur? Yes, no; describe |
| **Follow-up visit** | |
| • Date | • Date |
| • Alive fetus: yes; no | • Alive fetus: yes; no |

(*Continued*)

**Table 3.** (Continued)

| Paris (Smart FETO) | Leuven (Smart Removal) |
|---|---|
| • Balloon in place yes; no | • Balloon in place yes; no |
| • Balloon length (mm) | • Balloon length (mm) |
| • Balloon width (mm) | • Balloon width (mm) |
| • Cervical length (mm) | • Cervical length (mm) |
| • Chorionic membrane separation | • Chorionic membrane separation |
| • Deepest vertical amniotic fluid pocket (cm) | • Deepest vertical amniotic fluid pocket (cm) |
| • Any additional findings: yes; no; describe | • Any additional findings: yes; no; describe |
| • Did any A/E/SAE occur? Yes, no; describe | • Did any A/E/SAE occur? Yes, no; describe |
| Did any device deficiency occur? Yes, no; describe | Did any device deficiency occur? Yes, no; describe |
| • **Method for LHR measurement: tracing; longest diameter; anteroposterior diameter and perpendicular** | |
| •O/E LHR (%) | O/E LHR (%) |
| **Balloon removal procedure** | |
| • Date for balloon removal | • Date for balloon removal |
| • Removal context: at scheduled date; at emergency (labor; PROM; other) | • Removal context: at scheduled date; at emergency (labor; PROM; other) |
| • Balloon in place yes; no | • Balloon in place yes; no |
| • **Balloon length (mm)** | |
| • **Balloon width (mm)** | |
| • Physician responsible for procedure | • Physician responsible for procedure |
| • Date and data related to ultrasound scan at removal, including balloon shape and measurements | • Date and data related to ultrasound scan at removal, including balloon shape and measurements |
| • MRI exposure: no; yes | • MRI exposure: no; yes |
| • **MRI type: 1 tesla; 1.5; 2; 3** | |
| • MRI Brand and model | • MRI Brand and model |
| • Ultrasound scan after first tour around MRI including balloon visualization, position, shape, and measurements | • Ultrasound scan after first tour around MRI including balloon visualization, position, shape, and measurements |
| • Successful deflation: no; yes | • Successful deflation: no; yes |
| • Ultrasound scan after additional tour around MRI including balloon visualization, position, **shape, and measurements** | • Ultrasound scan after additional tour around MRI for balloon visualization |
| • Balloon removal before birth: no; yes | • Balloon removal before birth: no; yes |
| • Time between unplug and birth less than 24h: no; yes | • Time between unplug and birth less than 24h: no; yes |
| • Route of removal: MRI; fetoscopic, ultrasound-guided puncture; on placental circulation; off placental circulation; spontaneous deflation | • Route of removal: MRI; fetoscopic, ultrasound-guided puncture; on placental circulation; off placental circulation; spontaneous deflation |
| **Ultrasound scan after balloon deflation** | |
| • Alive fetus: yes; no | • Alive fetus: yes; no |
| • Balloon in place yes; no | • Balloon in place yes; no |
| • Balloon length (mm) | • Balloon length (mm) |
| • Balloon width (mm) | • Balloon width (mm) |
| • Cervical length (mm) | • Cervical length (mm) |
| • Chorionic membrane separation | • Chorionic membrane separation |
| • Deepest vertical amniotic fluid pocket (cm) | • Deepest vertical amniotic fluid pocket (cm) |
| • Any additional findings: yes; no; describe | • Any additional findings: yes; no; describe |
| **Post balloon deflation visit** | |
| • Date | • Date |
| • Alive fetus: yes; no | • Alive fetus: yes; no |
| • Cervical length (mm) | • Cervical length (mm) |
| • Chorionic membrane separation | • Chorionic membrane separation |
| • Deepest vertical amniotic fluid pocket (cm) | • Deepest vertical amniotic fluid pocket (cm) |

*(Continued)*

**Table 3.** (Continued)

| Paris (Smart FETO) | Leuven (Smart Removal) |
|---|---|
| • Any additional findings: yes; no; describe | • Any additional findings: yes; no; describe |
| • Did any A/E/SAE occur? Yes, no; describe | • Did any A/E/SAE occur? Yes, no; describe |
| Did any device deficiency occur? Yes, no; describe | Did any device deficiency occur? Yes, no; describe |
| • **Method for LHR measurement: tracing; longest diameter; anteroposterior diameter and perpendicular** | |
| • O/E LHR (%) | • O/E LHR (%) |
| **Delivery** | |
| • Date of delivery | • Date of delivery |
| • Center of delivery | • Center of delivery |
| • Premature rupture of membranes: no; yes (date) | • Premature rupture of membranes: no; yes (date) |
| • Indication for birth: elective birth; spontaneous birth; fetal complications; maternal complications | • Indication for birth: elective birth; spontaneous birth; fetal complications; maternal complications |
| • Route of birth: vaginal; assisted vaginal; primary caesarean; secondary caesarean; on placental circulation. | • Route of birth: vaginal; assisted vaginal; primary caesarean; secondary caesarean; on placental circulation. |
| • Liveborn: yes; no | • Liveborn: yes; no |
| • Gender: male; female | • Gender: male; female |
| • Birthweight (g) | • Birthweight (g) |
| • Apgar at 1' | • Apgar at 1' |
| • Apgar at 5' | • Apgar at 5' |
| • Arterial pH | • Arterial pH |
| • **Balloon visualization at birth: no; yes (shape, localization, need for withdrawal and modalities)** | • **Postpartum ultrasound performed: yes; no** |
| | • **Date of ultrasound** |
| | • **Balloon location: not visible; intracavitary; intramural; cervical; intraabdominal** |
| | • **Balloon location: placenta/membranes; amniotic fluid; uterus; newborn airways; newborn gastrointestinal tract; not visible; other (specify).** |
| • Did any A/E/SAE occur? Yes, no; describe | • Did any A/E/SAE occur? Yes, no; describe |
| **Neonatal outcome** | |
| • Postnatal death | • Postnatal death |
| • Cause of death | • Cause of death |
| | • Tracheal diameter at thoracic entry (mm) |
| | • Tracheal diameter 10 mm above the carina (mm) |
| | • Tracheal diameter at mid-distance (mm) |
| • Signs of symptoms of tracheomalacia: yes; no (specify) | • Signs of symptoms of tracheomalacia: yes; no (specify) |
| • Did any A/E/SAE occur? Yes, no; describe | • Did any A/E/SAE occur? Yes, no; describe |
| • Date of discharge from the NICU | • Date of discharge from the NICU |
| • **Survival at day 28, 56, at discharge from NICU, at discharge from Hospital, at 6 months: no; yes** | |
| • **Date of discharge from Hospital** | |
| • **Date of surgery** | |
| • **Use of patch at surgery: no; yes** | |
| • **Defect size: A,B,C,D** | |
| • **Occurrence of pulmonary hypertension: no; yes** | |
| • **Date of onset pulmonary hypertension** | |
| • **Need for oxygen at day 28 and 56: no; yes** | |
| • **Grade of oxygen dependency at day 28 and 56: 0 (no BPD); I (FiO2 21% or room air); II (FiO2 22•29%); III (FiO2 >29%, CPAP or mechanical ventilation)** | |
| • **Need for ECMO: no; yes** | |

(Continued)

**Table 3.** (Continued)

| Paris (Smart FETO) | Leuven (Smart Removal) |
|---|---|
| • **Duration of ECMO (days)** | |
| • **Total duration of ventilatory support (days)** | |
| • **Age at full enteral feeding (days)** | |
| • **Periventricular leukomalacia: no; grade I, grade II; grade III; not applicable** | |
| • **Intraventricular hemorrhage: no; grade I, grade II; grade III; not applicable** | |
| • **Sepsis: no; yes; not applicable** | |
| • **Necrotizing enterocolitis: no; yes; not applicable** | |
| • **Retinopathy of prematurity: yes (grade III or higher); no (<grade III); not applicable** | |
| • **Presence of reflux (>1/3 of the esophagus on clinically indicated radiography); no; yes; not applicable** | |
| • **Treatment of reflux: none; medical; surgical; other** | |
| • **Oxygen at discharge from Hospital: no; yes (grade of oxygen dependency)** | |
| • **Oxygen at 6 months: no; yes (grade of oxygen dependency)** | |

Abbreviations: LHR; lung-to-head ratio: O/E; observed-to-expected; w; weeks; CDH; congenital diaphragmatic hernia; FETO, Fetoscopic Endoluminal Tracheal Occlusion; MRI, magnetic resonance image; A/E, adverse event; SAE, serious adverse event; NICU; neonatal intensive care unit; ECMO, extracorporeal membrane oxygenation; PROM, premature rupture of the membranes; EXIT, ex-utero intrapartum treatment.

**Reestablishment of the fetal airways.** The Smart-TO deflation protocol is displayed in S1 Video.

- The patient is positioned in front of the MRI, her abdomen facing the front of the tunnel of the machine.

- The patient walks (or is strolled) around the machine while staying as close as possible to the machine.

- When approaching the rear of the tunnel, the patient positions herself in the middle of it, facing the tunnel and makes a short stop.

- Then she continues to walk (or being strolled) around the MRI while staying as close as possible to the machine

- Once she has completed the turn, she can leave the MRI room.

- Ultrasound is then performed independently by two experienced sonographers, to assess balloon deflation. When inflated, the balloon is easily visible on ultrasound as an anechoic structure. Balloon deflation will be indicated by visualization of the balloon on ultrasound before MRI exposure and its disappearance immediately after MRI exposure. In the case of deflation failure, a second or third MRI exposure will be attempted, again followed by ultrasound confirmation of balloon deflation.

**Conventional reestablishment of the fetal airways.** In the case of failure to deflate, balloon removal will be done as currently done with the conventional balloon, either ultrasound-guided puncture, fetoscopy, or in an emergency during abdominal delivery while the fetus is on placental circulation, or after birth by puncture above the manubrium sterni [14].

| TIMEPOINT | Enrolment | Post-allocation | | | | | | Close-out |
|---|---|---|---|---|---|---|---|---|
| | $-t_1$ | FETO* | Day 1 Post FETO | Weekly follow up | Unplug^ | Weekly follow up | Delivery and neonatal management | $t_x$ |
| **ENROLMENT** | | | | | | | | |
| Eligibility screen | X | | | | | | | |
| Informed consent | X | | | | | | | |
| Fetal ultrasound | X | X | X | X | X | X | | |
| **INTERVENTIONS** | | | | | | | | |
| *Hospitalization* | | ●———● | | | | | X | |
| *Placement of Smart-TO* | | X | | | | | | |
| *MRI exposure* | | | | | X | | | |
| *X-ray newborn* | | | | | | | X | |
| *Postpartum ultrasound uterus if indicated* | | | | | | | X | |
| **ASSESSMENTS** | | | | | | | | |
| *O/E LHR* | X | | | X | | | | |
| *Reporting of side effects* | | ●———————————————● | | | | | | |
| *Balloon deflation after MRI* | | | | | X | | | |
| *Balloon expulsion and localization* | | | | | | | X | |

**Fig 1. SPIRIT schedule.** Abbreviations: FETO, Fetoscopic Endoluminal Tracheal Occlusion; O/E LHR, observed-to-expected lung-to-head ratio; MRI, Magnetic Resonance Image.

## Ethics statement

In France, approval was provided by the committee for the protection of persons concerned (CPP "Ile de France VIII") in January 2021 (# 21 01 01), and the French medicines controls authorities (ANSM) in March (2020-A02834-35-A). In Belgium, approval was given by the Ethics Committee on Clinical Studies of the University Hospitals Leuven in July 2021 (S65423). The study was registered at the Federal Agency for Medicines and Health Products (FAGG/80M0892).

Written informed consent will be obtained from all participants.

## Discussion

Based on robust clinical evidence, one should consider the option of FETO in selected fetuses with CDH [2–5]. One of the major concerns about FETO is the potential problems related to balloon removal [9]. The Smart-TO balloon addresses this issue by allowing a noninvasive, easily triggered, and externally controlled reversal of occlusion [10]. After extensive translational research, the time has come to assess the efficacy of reversal of the occlusion and the safety of this new device in a first-in-woman study.

The main objective of this study is to demonstrate the ability to successfully deflate the Smart-TO balloon by the magnetic fringe field generated by an MRI scanner. The present trial also aims to demonstrate the Smart-TO balloon is no longer within the airways. Non-visualization of the balloon will provide evidence for airway permeability. In the Belgian site, the E.C. also required to positively identify the localization of the balloon following deflation, either within the amniotic fluid, membranes, or placenta (at delivery), and exclude its persistence in the uterus by postpartum ultrasound. Additional objectives of this study include the evaluation of safety, even though no serious adverse effects directly related to the Smart-TO balloon are anticipated.

The dimensions of the Smart-TO balloon and material (latex) are the same as the Goldbal2® balloon. For this reason, it is anticipated that the Smart-TO will induce similar lung growth compared to the Goldbal2® balloon, as previously demonstrated in preclinical studies [11, 15]. Additional outcome measurements include the occurrence of membrane rupture and preterm delivery, which is consistently reported in all FETO series. We will also report on the consequence of the above.

The limitations of our study will be that this is a non-comparative trial. However, including a second arm, where controls would have FETO by means of the Goldbal2® balloon, appears to be unethical, since this will not provide new data and there is sufficient data on file on outcomes when using the standard balloon.

In conclusion, this first in-woman study aims to demonstrate the ability of Smart-TO balloon to be prenatally deflated by the magnetic fringe field generated by an MRI scanner, its expulsion from the airways, as well as the safety of its use.

## Supporting information

**S1 Video.**
(MP4)

**S1 File. SPIRIT 2013 checklist: Recommended items to address in a clinical trial protocol and related documents**∗**.**
(DOC)

**S1 Text. Clinical trial protocol.**
(DOCX)

## Acknowledgments

The Smart-TO balloon is the result of several years of research and development within a consortium of the following institutions: BS-Medical Tech Industry (manufacturer), Strasbourg University Hospital, University of Strasbourg, INSERM Unit 1121 "Biomaterials and Bioengineering", Institut Hospitalo-Universitaire de Strasbourg, Institute for Research Against Cancers of the Digestive System, SATT-Conectus Alsace, Simian Laboratory Europe, and the KU Leuven.

The authors thank URC-CIC Necker Cochin (Adèle Bellino), the DRCI (Shoreh Azimi) and UZ Leuven Clinical Trial Centre (Veerle Doozen) for the implementation, monitoring and data management of the study.

## Author Contributions

**Conceptualization:** Nicolas Sananès, David Basurto, Anne-Gaël Cordier, Francesca Maria Russo, Alexandra Benachi, Jan Deprest.

**Funding acquisition:** Jan Deprest.

**Methodology:** David Basurto, Anne-Gaël Cordier, Caroline Elie, Francesca Maria Russo, Alexandra Benachi.

**Supervision:** Alexandra Benachi, Jan Deprest.

**Writing – original draft:** Nicolas Sananès, David Basurto, Anne-Gaël Cordier, Alexandra Benachi, Jan Deprest.

**Writing – review & editing:** Nicolas Sananès, Anne-Gaël Cordier, Francesca Maria Russo, Alexandra Benachi, Jan Deprest.

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
