## [Decision Letter · Decision Letter 0]

6 Nov 2022

PONE-D-22-21857Fetoscopic Endoluminal Tracheal Occlusion with Smart-TO balloon: efficacy of removal and safetyPLOS ONE

Dear Dr. Sananès,

Thank you for submitting your manuscript to PLOS ONE. After careful consideration, we feel that it has merit but does not fully meet PLOS ONE’s publication criteria as it currently stands. Therefore, we invite you to submit a revised version of the manuscript that addresses the points raised by all reviewers during the review process.

We look forward to receiving your revised manuscript.

Kind regards,

Abhishek Makkar, M.D.

Academic Editor

PLOS ONE

Journal Requirements:

"Nicolas Sananès is the primary co-inventor of the Smart-TO balloon. None of the authors has any financial interest in BS‐Medical Tech Industry, manufacturing the balloon. There are no other conflicts of interest."

6. We note that the original protocol that you have uploaded as a Supporting Information file contains an institutional logo. As this logo is likely copyrighted, we ask that you please remove it from this file and upload an updated version upon resubmission.

Reviewers' comments:

Reviewer's Responses to Questions

**Comments to the Author**

1. Does the manuscript provide a valid rationale for the proposed study, with clearly identified and justified research questions?

Reviewer #1: No

Reviewer #2: Yes

Reviewer #3: Yes

2. Is the protocol technically sound and planned in a manner that will lead to a meaningful outcome and allow testing the stated hypotheses?

Reviewer #1: No

Reviewer #2: Yes

Reviewer #3: Yes

3. Is the methodology feasible and described in sufficient detail to allow the work to be replicable?

Reviewer #1: No

Reviewer #2: Yes

Reviewer #3: Yes

4. Have the authors described where all data underlying the findings will be made available when the study is complete?

Reviewer #1: No

Reviewer #2: Yes

Reviewer #3: Yes

5. Is the manuscript presented in an intelligible fashion and written in standard English?

Reviewer #1: Yes

Reviewer #2: Yes

Reviewer #3: Yes

6. Review Comments to the Author

You may also provide optional suggestions and comments to authors that they might find helpful in planning their study.

Reviewer #1: The authors reported the study protocol of the single arm phase I trial conducted in parallel in Paris and Leuven about the use of the Smart-TO balloon for congenital diaphragmatic hernia repair. They concluded, that this phase-I study may provide the first evidence of the potential to reverse the occlusion by Smart-TO and free the airways non-invasively, as well as safety data.

There remain some questions regarding the protocol, which I would like to ask the authors.

1. The title should be added by the fact, that the paper is about the study protocol rather than study results.

2. I am a bit sceptic about the classification as Phase 1. The term first in human (patients) trial clearly describes what will be done.

3. As these first in human trials usually will focus on safety (see Guideline on strategies to identify and mitigate risks for first-in-human and early clinical trials with investigational medicinal products, EMA) the objective of the study should be refined from efficacy to safety. Here special focus should be paid to the pediatric and rare disease nature of the disease.

4. I wonder myself - as this is somewhat unclear - a sequential recruitment strategy with continuous monitoring for safety applied combined in both locations, would reflect safety aspects more appropriate as a fixed sample. Other designs, which might be considered is Simon's two Stage design.

For both: (http://cancer.unc.edu/biostatistics/program/ivanova/ContinuosMonitoringForToxicity.aspx).

5. As the number of subjects is rather high for safety and differs in both locations, I think even this should be refined. Numbers of up to ten in sequential order administration might be reasonable.

Please note, that even if efficacy would be allowed as study endpoint, the effect size should be the same in both locations, as the trial is the same , and thus the sample size should be the same.

6. I wonder why the study must be conducted parallel in Paris and in Leuven. The first in human trial aims could be answered in one center, right?

7. The sampling plan, in particular what is written in "Existing data", "Recruitement" and "Sample size" needs review, as the delivered text does not meet the required information, or in the case of sample size is not appropriate to FiH trials.

8. Co-Primary endpoint in the Paris Trial need a statistical decision rule and considerations about the type one error inflation.

9. The parallel conduct of the trial needs a clear decision rule about the potential / desired outcome of the trial.

Reviewer #2: I REALLY enjoyed reading your manuscript. This is incredibly important work. Thank you

Below you will find some thoughts/comments/questions that I had while reading your manuscript.

Page 12 – line 2 - Reversal is at present an invasive procedure that can be performed by either ultrasound-guided puncture, fetoscopy, or, less ideal, while the fetus is maintained on placental circulation or at birth after vaginal delivery [10].

From the paper cited “Primary attempt was by fetoscopy in 196 (67.1%), by ultrasound-guided puncture in 62 (21.2%), by tracheoscopy on placental circulation in 30 (10.3%), and postnatal tracheoscopy in 4 cases (1.4%).

Would recommend re-wording the sentence to something like this “Reversal is at present an invasive procedure that can be performed prenatally by either ultrasound-guided puncture, fetoscopy, or, less ideal, after delivery of the baby prior to cord clamping while the fetus is maintained on placental circulation or after the cord is clamped at birth after vaginal delivery [10].

Page 12 – “The only neonatal deaths that occurred, were when balloon reversal was attempted in centers without experience or that were unprepared [10].” I think this should be worded stronger …. 3/9 babies that underwent a postnatal balloon removal died. These were in non-FETO centers. However, even in FETO centers it is difficult to remove the balloon postnatally. 1/3 are removed emergently prenatally and in the post-natal situation, there is a lot of (published and unpublished) morbidity concerns. This is VERY IMPORTANT! The concept of magnetic balloon removal prior to delivery to avoid all of this is – is SO, SO important!!

Page 12 – “In conclusion, the occlusion period is a serious burden on patients who are requested to stay close to the FETO center until balloon removal, as well as for the fetal surgery centers because of the need for permanently available staff. All these conditions, limit the acceptability of FETO as being practiced today.”

An experienced FETO team available 24/7 is required and even with this team, balloon removal can be difficult. Plus, it is not just about balloon removal – also management of the secretions and plugs if balloon is removed emergently in the postnatal situation (either before or after cord is clamped). Lots of potential morbidity. This balloon is the answer to SO many concerns, fears, problems that have happened. Again, I think the wording should be stronger here.

Page 13 – you mention different positions and height, but does the degree of polyhydramnios affect the magnetic affect?

Page 13 – the proximity to the MRI will cause the balloon to 1. deflate, then 2. to be expelled, correct? How do you know that the MRI does not cause the balloon to be expelled while inflated? If this is theoretically possible, maybe vocal cord injury and airway injury should be assessed with a post-natal ENT evaluation

Page 13 “Other objectives include assessment of prematurity, preterm premature rupture

of membranes, lung growth, neonatal survival, and the need for oxygen supplementation at

discharge from the hospital.”

Should you also assess ventilator days, need for tracheostomy (related to vocal cord injury as well as chronic lung disease), need for ECMO, length of stay and discharge on pulmonary hypertension medications

Page 16 – “assessed through ultrasound immediately after MRI exposure”

Have you done ultrasound during the MRI exposure? (to ensure balloon is deflated first … )

Page 18 – “In the Belgian site, the E.C. also required to positively identify the localization of the balloon following deflation” – this makes sense – also to make sure balloon is not expelled into the mouth and then with first breath at delivery could migrate to airway or esophagus (probably impossible, but thought to mention as it crossed my mind)

Page 23 – “Balloon expulsion from the fetal airways (by postnatal chest X-ray)” – the balloon is radio-opaque? This was not clear to me in the study protocol

Page 25 – should it be stated US by trace or AP method? MRI by Myers or Rypens method?

Page 28 – neonatal outcomes – include stridor? aspiration? tracheostomy? (secondary to vocal cord injury/immobility/paralysis)

Page 28 – would be nice if both hospitals measured tracheal diameter postnatally and required formal neonatal ENT evaluation

Page 29 – not sure all of these outcomes are necessary (?) …. NEJM publication showed balloon impacts these outcomes. I would like to see more on description of pulmonary secretions/plugs at birth, first blood gas, first CXR (open or bilateral opacification due to airway secretions/plugs), time to intubation after delivery, neonatal …. In addition to how many required emergent removal at the time of delivery either before or after cord is clamped – this is really the most important feature

Reviewer #3: Thank you for allowing me the opportunity to review your work. Please see some of my comments below

Page 11, 3rd paragraph

"A second disadvantage of the current procedure is the need for a second intervention

to reverse the occlusion and re-establish airway patency." Consider wording this as "Another (there are a few disadvantages) disadvantage of the current procedure is the need for an invasive, second intervention to reverse..." I think the possibility of a non-invasive intervention is critical.

Page 12,

"The only neonatal deaths that occurred, were when balloon reversal was attempted in centers without experience or that were unprepared [10]." Consider emphasizing this- neonatal deaths occurred only at non-FETO centers. 3 of 9 (33%) interventions done outside FETO centers led to neonatal death, presumably due to lack of technical expertise. This non-invasive procedure might offer better outcomes when families are unable to return to the FETO center.

Page 13,

"In that experiment, deflation was successfully achieved regardless of the fetal position and the exact level of the fetus from the ground [12]" Would you consider elaborating more on this? The ex-vivo findings in the standing and "lying-on-a-stretcher" were very reassuring. The only failed deflation was for the mother sitting in a wheelchair.

"In the latter experiment, fetal lambs expelled the Smart-TO balloon following exposure to the fringe field of a 3T MRI."

Forgive me if I misread this, but for deflation (albeit in an experimental environment) a 1.5T MRI was sufficient, however the expulsion occurred in the field of a 3T MRI? Does this need to be clarified? Will both deflation and expulsion be done around a standard 1.5T MRI? Also are there any concerns for airway damage in the setting of the expulsion of a partially deflated balloon?

"The main objective of the study is to demonstrate the ability to consistently deflate the balloon prenatally by the magnetic fringe field generated by a clinical MRI scanner, and that it will be expelled from the airway" Are there two primary objectives? Deflation AND expulsion? Also, is it essential to clarify if this will be a 1.5T or 3T MRI?

Page 15,

"... hypothesized that for a 100% deflation and expelling rate". Consider using deflation and expulsion rate

Page 23 Neonatal secondary endpoints (Belgium)

"Tracheal diameter on first postnatal chest X-ray". Is there currently a standard method for this measure (vertebral level etc.)? Also, is this affected by size of endotracheal tube?

"Assessment for any local side effects of the balloon (signs or symptoms of tracheomegaly and /or tracheomalacia)" At what age will this be done? Will tracheomalacia be diagnosed via DLB?

7. PLOS authors have the option to publish the peer review history of their article (what does this mean?). If published, this will include your full peer review and any attached files.

Reviewer #1: No

Reviewer #2: No

Reviewer #3: **Yes: **Vedanta S. Dariya

---

## [Author Response · Author response to Decision Letter 0]

27 Dec 2022

PONE-D-22-21857

Fetoscopic Endoluminal Tracheal Occlusion with Smart-TO balloon: efficacy of removal and safety

PLOS ONE

Dear Editor, dear reviewers, 

We would like to thank you for considering our protocol paper and for your precious comments. The manuscript was revised along those lines and editing modifications were made according to journal requirements. 

Some suggested modifications were not possible since the trial protocol was approved by the relevant regulatory bodies and is now recruiting.

We hope that this version will be considered of scientific interest to be published in Plos One.

Sincerely yours, on behalf of the authors,

Nicolas Sananès

Journal Requirements:

All these requirements have been addressed.

The manuscript has been modified in order to meet PLOS ONE’s style requirements. File naming has been updated as well.

"Nicolas Sananès is the primary co-inventor of the Smart-TO balloon. None of the authors has any financial interest in BS‐Medical Tech Industry, manufacturing the balloon. There are no other conflicts of interest."

The competing interest statement has been modified in the manuscript and has been added to the cover letter.

That section now reads:'Nicolas Sananès is the primary co-inventor of the Smart-TO balloon. None of the authors has any financial interest in BS‐Medical Tech Industry, manufacturing the balloon. There are no other conflicts of interest. This does not alter our adherence to PLOS ONE policies on sharing data and materials.’

The ORCID iD has been updated and validated.

The ethics statement now appears at the end of the Methods section.

5. Please include captions for your Supporting Information files at the end of your manuscript, and update any in-text citations to match accordingly. Please see our Supporting Information guidelines for more information: http://journals.plos.org/plos one/s/supporting-information.

Supporting information captions have been added at the end of the manuscript.

6. We note that the original protocol that you have uploaded as a Supporting Information file contains an institutional logo. As this logo is likely copyrighted, we ask that you please remove it from this file and upload an updated version upon resubmission.

We removed institutional logos from protocols.

Reviewers' comments:

Reviewer #1: The authors reported the study protocol of the single arm phase I trial conducted in parallel in Paris and Leuven about the use of the Smart-TO balloon for congenital diaphragmatic hernia repair. They concluded, that this phase-I study may provide the first evidence of the potential to reverse the occlusion by Smart-TO and free the airways non-invasively, as well as safety data.

There remain some questions regarding the protocol, which I would like to ask the authors.

1. The title should be added by the fact, that the paper is about the study protocol rather than study results.

Thank you, we modified the title in this way.

The title now reads: ‘Fetoscopic Endoluminal Tracheal Occlusion with Smart-TO balloon: study protocol to evaluate effectiveness and safety of non-invasive removal.’

2. I am a bit sceptic about the classification as Phase 1. The term first in human (patients) trial clearly describes what will be done.

We made this modification.

The text has been modified to: ‘This first in human trial may provide the first evidence of the potential to reverse the occlusion by Smart-TO and free the airways non-invasively, as well a safety data.’

3. As these first in human trials usually will focus on safety (see Guideline on strategies to identify and mitigate risks for first-in-human and early clinical trials with investigational medicinal products, EMA) the objective of the study should be refined from efficacy to safety. Here special focus should be paid to the pediatric and rare disease nature of the disease.

Thank you. We agree that this is a safety study, during which effective deflation and expelling is being tested, while monitoring other potential fetal or maternal complications. Using efficacy of deflation was a misleading wording so we changed for effective deflation, which is indeed a part of the safety evaluation.

The abstract has been modified to: ‘Our main objective is to evaluate the effectiveness of prenatal deflation of the balloon by the magnetic field generated by an MRI scanner.’

We also changed the title in this way (see above in 1).

4. I wonder myself - as this is somewhat unclear - a sequential recruitment strategy with continuous monitoring for safety applied combined in both locations, would reflect safety aspects more appropriate as a fixed sample. Other designs, which might be considered is Simon's two Stage design.

For both: (http://cancer.unc.edu/biostatistics/program/ivanova/ContinuosMonitoringForToxicity.aspx).

Thank you for this suggestion. However, according to national regulations both studies are run under responsibility and control of the two institutions, and are approved and considered as two separate studies. Both studies were deemed to be independent and powered by the Ethics Committees or its equivalent, for the research question asked. 

5. As the number of subjects is rather high for safety and differs in both locations, I think even this should be refined. Numbers of up to ten in sequential order administration might be reasonable.

Please note, that even if efficacy would be allowed as study endpoint, the effect size should be the same in both locations, as the trial is the same , and thus the sample size should be the same.

6. I wonder why the study must be conducted parallel in Paris and in Leuven. The first in human trial aims could be answered in one center, right?

As above; the study was powered for the key safety parameter, i.e. deflation and expelling from the airways (without further neonatal outcome). Both studies were powered for that, and again, in two local independent cohort.

7. The sampling plan, in particular what is written in "Existing data", "Recruitement" and "Sample size" needs review, as the delivered text does not meet the required information, or in the case of sample size is not appropriate to FiH trials.

We’re sorry but we’re not sure to understand your point. At the same time, we can not change the recruitment and sample size calculation anymore since the trial protocol was approved by the relevant regulatory bodies and is now recruiting.

8. Co-Primary endpoint in the Paris Trial need a statistical decision rule and considerations about the type one error inflation.

There is a single endpoint in the Paris trial but the phrasing was misleading. We rephrased this part and better explain the differences between the two protocols regarding the primary endpoint, without co-primary outcome.

The modified section reads as follows: 

‘In Paris, the primary endpoint is the successful deflation of the Smart-TO balloon after exposure to the fringe field of the MRI, assessed through ultrasound immediately after MRI exposure and the expulsion of the Smart-TO balloon from the airways, as documented by a X-ray of the neonatal chest at birth (the valve of the balloon is radio-opaque).

In Leuven, only the successful deflation of the Smart-TO balloon after exposure to the fringe field of the MRI is the efficacy endpoint. The latter endpoint will be therefore be considered as the common primary endpoint. For Leuven, the expulsion of the Smart-TO balloon from the airways is considered as a secondary endpoint.’

9. The parallel conduct of the trial needs a clear decision rule about the potential / desired outcome of the trial.

There is no prespecified early stopping rule, but every serious adverse effect is urgently reported to an Data Safety Monitoring Board, which can decide to stop the study.

 

Reviewer #2: I REALLY enjoyed reading your manuscript. This is incredibly important work. Thank you

Thank you for this comment.

Below you will find some thoughts/comments/questions that I had while reading your manuscript.

Page 12 – line 2 - Reversal is at present an invasive procedure that can be performed by either ultrasound-guided puncture, fetoscopy, or, less ideal, while the fetus is maintained on placental circulation or at birth after vaginal delivery [10].

From the paper cited “Primary attempt was by fetoscopy in 196 (67.1%), by ultrasound-guided puncture in 62 (21.2%), by tracheoscopy on placental circulation in 30 (10.3%), and postnatal tracheoscopy in 4 cases (1.4%).

Would recommend re-wording the sentence to something like this “Reversal is at present an invasive procedure that can be performed prenatally by either ultrasound-guided puncture, fetoscopy, or, less ideal, after delivery of the baby prior to cord clamping while the fetus is maintained on placental circulation or after the cord is clamped at birth after vaginal delivery [10].

Thank you, we made the change following your recommendation: ’ Reversal is at present an invasive procedure that can be performed prenatally by either ultrasound-guided puncture, fetoscopy, or, less ideal, after delivery of the baby prior to cord clamping while the fetus is maintained on placental circulation or after the cord is clamped at birth after vaginal delivery [10].’

Page 12 – “The only neonatal deaths that occurred, were when balloon reversal was attempted in centers without experience or that were unprepared [10].” I think this should be worded stronger …. 3/9 babies that underwent a postnatal balloon removal died. These were in non-FETO centers. However, even in FETO centers it is difficult to remove the balloon postnatally. 1/3 are removed emergently prenatally and in the post-natal situation, there is a lot of (published and unpublished) morbidity concerns. This is VERY IMPORTANT! The concept of magnetic balloon removal prior to delivery to avoid all of this is – is SO, SO important!!

Page 12 – “In conclusion, the occlusion period is a serious burden on patients who are requested to stay close to the FETO center until balloon removal, as well as for the fetal surgery centers because of the need for permanently available staff. All these conditions, limit the acceptability of FETO as being practiced today.”

An experienced FETO team available 24/7 is required and even with this team, balloon removal can be difficult. Plus, it is not just about balloon removal – also management of the secretions and plugs if balloon is removed emergently in the postnatal situation (either before or after cord is clamped). Lots of potential morbidity. This balloon is the answer to SO many concerns, fears, problems that have happened. Again, I think the wording should be stronger here.

Thank you and indeed we agree… We added a stronger wording in the introduction section when introducing the Smart-TO balloon.

This following part has been added : ‘Therefore, the Smart-TO balloon may address issues related to the unplug procedure, i.e. neonatal deaths by failure of balloon removal, morbidity related to a second fetal surgery procedure, need for FETO centers with experienced team available 24/7, and need for the pregnant women to stay close to a FETO center during the whole duration of the occlusion.’

Page 13 – you mention different positions and height, but does the degree of polyhydramnios affect the magnetic affect?

We were not able to assess this point in preclinical tests but don’t anticipate that polyhydramnios affect the magnetic effect since it may not significantly increase distance between balloon and MRI. Moreover, it’s not only a matter of strength of the magnetic field but also a matter of modification of the polarization while the patient turns around the MRI. 

Page 13 – the proximity to the MRI will cause the balloon to 1. deflate, then 2. to be expelled, correct? How do you know that the MRI does not cause the balloon to be expelled while inflated? If this is theoretically possible, maybe vocal cord injury and airway injury should be assessed with a post-natal ENT evaluation

During inflation, the balloon, as its predecessor, should stay in place. While being deflated its diameter will decrease, a process that happens in seconds. The suggested trauma related to the migration of the balloon cannot be excluded but is theoretical : (1) no such observations were made in preclinical experiments ; (2) the insertion process with a rigid endoscope is far more traumatic ; (3) the suggested routine postnatal evaluation is an additional risk that is not justified by the information that would be retrieved. Tracheobronchoscopy would be done as clinically indicated.

Page 13 “Other objectives include assessment of prematurity, preterm premature rupture of membranes, lung growth, neonatal survival, and the need for oxygen supplementation at discharge from the hospital.”

Should you also assess ventilator days, need for tracheostomy (related to vocal cord injury as well as chronic lung disease), need for ECMO, length of stay and discharge on pulmonary hypertension medications

Thank you for the suggestion. These are all secondary outcome measurements, and the study is not powered for these. Apart from tracheal problems, this list is related to the effect of the occlusion and not specifically to the new balloon.

Page 16 – “assessed through ultrasound immediately after MRI exposure”

Have you done ultrasound during the MRI exposure? (to ensure balloon is deflated first … )

Patients are followed on a regular basis and would have ultrasound imaging prior to the planned MRI exposure. On ultrasound the balloon can be well visualized. Ultrasound scanning obviously cannot be performed during MRI exposure; this would be very dangerous.

Page 18 – “In the Belgian site, the E.C. also required to positively identify the localization of the balloon following deflation” – this makes sense – also to make sure balloon is not expelled into the mouth and then with first breath at delivery could migrate to airway or esophagus (probably impossible, but thought to mention as it crossed my mind)

Positively identify the localization of the balloon following deflation is indeed of interest but not the primary purpose. Still, the crucial point is that the balloon is expelled out of the airways, which can be demonstrated on neonatal X-ray since the valve of the balloon is radio-opaque. There is ample experience with the standard balloon on the fate of the balloon after puncture, i.e. typically migration to the stomach or amniotic cavity. 

Page 23 – “Balloon expulsion from the fetal airways (by postnatal chest X-ray)” – the balloon is radio-opaque? This was not clear to me in the study protocol

Indeed the balloon is radio-opaque. We clarified this point in the protocol.

Page 25 – should it be stated US by trace or AP method? MRI by Myers or Rypens method?

Yes, lung size measurement can be performed either by trace or anteroposterior diameters of the contralateral lung. We added this detail in the text. However, inclusion criteria do not include lung volume assessed by MRI, neither were they in the TOTAL trials.

Page 28 – neonatal outcomes – include stridor? aspiration? tracheostomy? (secondary to vocal cord injury/immobility/paralysis)

Page 28 – would be nice if both hospitals measured tracheal diameter postnatally and required formal neonatal ENT evaluation

Page 29 – not sure all of these outcomes are necessary (?) …. NEJM publication showed balloon impacts these outcomes. I would like to see more on description of pulmonary secretions/plugs at birth, first blood gas, first CXR (open or bilateral opacification due to airway secretions/plugs), time to intubation after delivery, neonatal …. In addition to how many required emergent removal at the time of delivery either before or after cord is clamped – this is really the most important feature

In terms of neonatal outcomes, the variables chosen are identical as in the TOTAL trials. The data from that paper will be used as a benchmark, and that is why we report the same outcomes, even if we don’t expect differences. The Smart-TO balloon has the same dimensions and is made of the same material as the Goldbal2 balloon.

We did not plan to systematically collect some of the above suggested neonatal outcomes, as they are quite descriptive. We have taken an approach to not use any restrictive definition of adverse effects, hence all will be reported (Table 3). 

As to the tracheal side effects, we refer to our earlier work in the sheep model focusing logically demonstrating similar tracheal side effects as the Goldbal 2 balloon (Basurto, D., et al., Safety and efficacy of the Smart Tracheal Occlusion device in the diaphragmatic hernia lamb model. Ultrasound Obstet Gynecol, 2020).

Thank you for your suggestion to measure tracheal diameters, which can be done. The geometrical effects of balloon occlusion have already been well described, and we expect similar effects. We we should be able to perform these measurements since neonatal X-ray is systematically performed. 

Reviewer #3: Thank you for allowing me the opportunity to review your work. Please see some of my comments below

Page 11, 3rd paragraph

"A second disadvantage of the current procedure is the need for a second intervention to reverse the occlusion and re-establish airway patency." Consider wording this as "Another (there are a few disadvantages) disadvantage of the current procedure is the need for an invasive, second intervention to reverse..." I think the possibility of a non-invasive intervention is critical.

Thank you, we made the modification following your recommendation : ‘Another disadvantage of the current procedure is the need for an invasive, second intervention to reverse the occlusion and re-establish airway patency.’

Page 12,

"The only neonatal deaths that occurred, were when balloon reversal was attempted in centers without experience or that were unprepared [10]." Consider emphasizing this- neonatal deaths occurred only at non-FETO centers. 3 of 9 (33%) interventions done outside FETO centers led to neonatal death, presumably due to lack of technical expertise. This non-invasive procedure might offer better outcomes when families are unable to return to the FETO center.

We added a stronger wording in the introduction section when introducing the Smart-TO balloon, by adding the following: ‘Therefore, the Smart-TO balloon may address issues related to the unplug procedure, i.e. neonatal deaths by failure of balloon removal, morbidity related to a second fetal surgery procedure, need for FETO centers with experienced team available 24/7, and need for the pregnant women to stay close to a FETO center during the whole duration of the occlusion.’

Page 13,

"In that experiment, deflation was successfully achieved regardless of the fetal position and the exact level of the fetus from the ground [12]" Would you consider elaborating more on this? The ex-vivo findings in the standing and "lying-on-a-stretcher" were very reassuring. The only failed deflation was for the mother sitting in a wheelchair.

We added details on this experiment: ‘In that experiment, deflation was successfully achieved using a 1.5T MRI in 100% of cases in a maternal standing position as well as when the maternal position was ‘lying on a stretcher’. The only case of failure occurred when the maternal position was ‘sitting in a wheelchair’, likely because of the distance between the MRI scanner and the patient in this scenario.’

"In the latter experiment, fetal lambs expelled the Smart-TO balloon following exposure to the fringe field of a 3T MRI."

Forgive me if I misread this, but for deflation (albeit in an experimental environment) a 1.5T MRI was sufficient, however the expulsion occurred in the field of a 3T MRI? Does this need to be clarified? Will both deflation and expulsion be done around a standard 1.5T MRI? Also are there any concerns for airway damage in the setting of the expulsion of a partially deflated balloon? 

Thank you, we expanded this. For the ex-vivo study, a 1.5T MRI was used and we made it clearer in the manuscript. The reason why a 3T was used for the sheep study is because it the MRI machine available in that animal lab. All other tests (including in vitro tests performed by the manufacturer) were performed with 1, 1.5 and 2T MRI. We included a video demonstrating the unplug procedure, in which we state that 1, 1.5 and 2T MRI can be used. To be noted that it’s not only a matter of strengthens of the magnetic field but also a matter of polarization and depolarization.

Regarding the risk of airway damage, in the introduction section we refer to a study we conducted in the sheep model focusing on tracheal side effects with histological analyses, which demonstrated similar tracheal side effects as the Goldbal 2 balloon (Basurto, D., et al., Safety and efficacy of the Smart Tracheal Occlusion device in the diaphragmatic hernia lamb model. Ultrasound Obstet Gynecol, 2020).

"The main objective of the study is to demonstrate the ability to consistently deflate the balloon prenatally by the magnetic fringe field generated by a clinical MRI scanner, and that it will be expelled from the airway" Are there two primary objectives? Deflation AND expulsion? Also, is it essential to clarify if this will be a 1.5T or 3T MRI?

Thank you, we clarified this point, stating that the primary endpoint is the deflation of the balloon.

The main objective was rephrased to make it clearer: ’ In France, the primary endpoint is the successful deflation of the Smart-TO balloon after exposure to the fringe field of the MRI, assessed through ultrasound immediately after MRI exposure and the expulsion of the Smart-TO balloon from the airways, as documented by a X-ray of the neonatal chest at birth (the valve of the balloon is radio-opaque).

In Leuven, only the successful deflation of the Smart-TO balloon after exposure to the fringe field of the MRI is required for efficacy; this endpoint will be then considered as the common primary endpoint. 

For Belgium, the expulsion of the Smart-TO balloon from the airways is considered as a secondary endpoint.’

Regarding the type of MRI, please see the answer above.

Page 15,

"... hypothesized that for a 100% deflation and expelling rate". Consider using deflation and expulsion rate

We understand it’s misunderstanding that the primary outcomes of the two parallel studies are different so we clarified this point. Still, we expect 100% deflation and expulsion rate.

Page 23 Neonatal secondary endpoints (Belgium)

"Tracheal diameter on first postnatal chest X-ray". 

Is there currently a standard method for this measure (vertebral level etc.)? Also, is this affected by size of endotracheal tube?

"Assessment for any local side effects of the balloon (signs or symptoms of tracheomegaly and /or tracheomalacia)" At what age will this be done? Will tracheomalacia be diagnosed via DLB?

This will be measured as previously described i.e. at the level of entry into the chest (mm), 1 cm above the carina (mm), and at mid-distance between these sites (mm). This is part of our routine follow up. 

Any tracheal symptoms may be reported (not-limitative list) but no invasive testing is done as this is not recommended for treatment with the current balloon, and seems not justified. 

Tracheomalacia has different definitions, but persistent symptomatic tracheomalacia is anyway uncommon complication (1%) and its occurrence cannot be determined in such a small study, let be its pathogenesis.

---

## [Editor Report · Decision Letter 1]

26 Jan 2023

Fetoscopic Endoluminal Tracheal Occlusion with Smart-TO balloon: study protocol to evaluate effectiveness and safety of non-invasive removal.

PONE-D-22-21857R1

Dear Dr. Sananès,

We’re pleased to inform you that your manuscript has been judged scientifically suitable for publication and will be formally accepted for publication once it meets all outstanding technical requirements.

Kind regards,

Abhishek Makkar, M.D.

Academic Editor

PLOS ONE
---

## [Editor Report · Acceptance letter]

3 Mar 2023

PONE-D-22-21857R1 

Fetoscopic Endoluminal Tracheal Occlusion with Smart-TO balloon: study protocol to evaluate effectiveness and safety of non-invasive removal. 

Dear Dr. Sananès:

I'm pleased to inform you that your manuscript has been deemed suitable for publication in PLOS ONE. Congratulations! Your manuscript is now with our production department. 

Kind regards, 

on behalf of

Dr. Abhishek Makkar 

Academic Editor

PLOS ONE